# Inverted formin 2 regulates intracellular trafficking, placentation, and pregnancy outcome

Katherine Young Bezold Lamm[1,2,3,4]*, Maddison L Johnson[5], Julie Baker Phillips[5], Michael B Muntifering[4], Jeanne M James[6], Helen N Jones[3], Raymond W Redline[7], Antonis Rokas[5], Louis J Muglia[1,2,8]*

[1]Center for the Prevention of Preterm Birth, Perinatal Institute, Cincinnati Children's Hospital Medical Center, Cincinnati, United States; [2]Department of Pediatrics, University of Cincinnati College of Medicine, Cincinnati, United States; [3]Molecular and Developmental Biology Graduate Program, University of Cincinnati College of Medicine, Cincinnati Children's Hospital Medical Center, Cincinnati, United States; [4]Division of Developmental Biology, Cincinnati Children's Hospital Medical Center, Cincinnati, United States; [5]Department of Biological Sciences, Vanderbilt University, Nashville, United States; [6]The Heart Institute, Cincinnati Children's Hospital Medical Center, Cincinnati, United States; [7]Department of Pathology, University Hospitals Cleveland Medical Center, Cleveland, United States; [8]Division of Human Genetics, Cincinnati Children's Hospital Medical Center, Cincinnati, United States

*For correspondence:
katie.bezold@cchmc.org (KYBL);
Louis.Muglia@cchmc.org (LJM)

**Abstract** Healthy pregnancy depends on proper placentation—including proliferation, differentiation, and invasion of trophoblast cells—which, if impaired, causes placental ischemia resulting in intrauterine growth restriction and preeclampsia. Mechanisms regulating trophoblast invasion, however, are unknown. We report that reduction of *Inverted formin 2* (*INF2*) alters intracellular trafficking and significantly impairs invasion in a model of human extravillous trophoblasts. Furthermore, global loss of *Inf2* in mice recapitulates maternal and fetal phenotypes of placental insufficiency. *Inf2*$^{-/-}$ dams have reduced spiral artery numbers and late gestational hypertension with resolution following delivery. *Inf2*$^{-/-}$ fetuses are growth restricted and demonstrate changes in umbilical artery Doppler consistent with poor placental perfusion and fetal distress. Loss of *Inf2* increases fetal vascular density in the placenta and dysregulates trophoblast expression of angiogenic factors. Our data support a critical regulatory role for *INF2* in trophoblast invasion—a necessary process for placentation—representing a possible future target for improving placentation and fetal outcomes.
DOI: https://doi.org/10.7554/eLife.31150.001

## Introduction

Implantation and placentation involve complex synchronization between the developing embryo and decidualization of the uterus. Extravillous trophoblasts (EVTs) differentiate from column cytotrophoblasts (CTBs), invade through the endometrium to the myometrium, and remodel decidual spiral arteries to form high-capacity, low-resistance vessels, supplying maternal blood to the lacunae surrounding the developing placental villi (*Damsky et al., 1992*; *Red-Horse et al., 2004*). Shallow invasion by EVTs and failed spiral artery remodeling yield peripheral vasoconstriction and high-resistance vessels thought to comprise the first stage of the development of preeclampsia (PE). Together with

**eLife digest** The placenta is an organ that develops with the baby during pregnancy and links the baby with his or her mother. This connection allows mom and baby to communicate throughout the pregnancy to share nutrition and growth signals, and to coordinate their immune systems. Abnormal placental growth can have lasting, harmful effects on the health of the mother and baby.

Specialized cells in the placenta called trophoblasts help the embryo implant into the mother's womb and direct the flow of nutrient- and oxygen-rich blood from the mother to the baby. If trophoblasts do not penetrate deeply enough into the womb, the mother will be at risk for developing a life-threating condition called preeclampsia, which occurs when her blood pressure becomes dangerously high. The only treatment is to deliver the baby. Her baby will also be at risk of poor growth and premature delivery. Scientists still do not know exactly how the trophoblasts invade the womb and what goes wrong that causes placental abnormalities.

Now, Lamm et al. show that losing a gene called *Inverted Formin 2*, or *Inf2* for short, which helps cells to form structures, causes placental abnormalities and preeclampsia symptoms in mice. In the experiments, trophoblasts in mice without *Inf2* were unable to invade the womb properly. The *Inf2*-lacking mice had fewer blood vessels feeding the placenta. These mice developed high blood pressure late in pregnancy, which returned to normal after their babies were born and the placentas expelled. During pregnancy, the placentas of *Inf2*-lacking mice were less efficient in transporting nutrients and gases, and their fetuses grew slowly and showed signs of distress.

This suggests that the *INF2* gene is necessary for the placenta to develop properly. Learning more about what can go wrong as the placenta forms might help physicians predict or prevent preeclampsia, fetal growth problems, and other placental abnormalities. More studies could determine if treatments targeting *INF2* would improve the development of the placenta, protect mothers from preeclampsia, and prevent conditions that slow down the babies' growth.

DOI: https://doi.org/10.7554/eLife.31150.002

reduced arterial compliance, these vascular changes result in the hypertensive phenotype characteristic of this disease (*Bosio et al., 1999*; *Wolf et al., 2001*).

The cause of shallow EVT invasion is unknown and under-investigated due to a lack of relevant animal models (*McCarthy et al., 2011*). For example, the reduced uterine perfusion pressure (RUPP) rat model of PE—which recapitulates systemic changes in maternal renal, immune, and circulatory functions—necessitates physical occlusion of the abdominal aorta and uterine arteries (*Alexander et al., 2001*; *Li et al., 2012*). The resulting placental ischemia begins at midgestation, well after artery remodeling. Several mouse models have been developed to understand the pathophysiology of this disease (*McCarthy et al., 2011*) such as the nitric oxide synthase knockout mouse (*Huang et al., 1993*; *Huang et al., 1995*; *Shesely et al., 2001*), the catechol-O-methyltransferase deficient mouse (*Kanasaki et al., 2008*), and the glial cells missing hypomorphic mouse (*Bainbridge et al., 2012*). These models recapitulate aspects of PE but the etiology of shallow EVT invasion, the early cause of placental ischemia, is still unknown.

Successful cellular invasion depends on formation of invasive structures such as invadopodia and podosomes (*Parast et al., 2001*; *Patel and Dash, 2012*), suggesting cytoskeletal integrity and appropriate remodeling is critical for EVT migration and invasion. Formins, a multi-domain family of proteins highly expressed in reproductive tissues (*Figure 1*), have been identified as critical in the regulation of cytoskeletal assembly and organization through actin polymerization and microtubule bundling, mediating processes such as cellular migration, division, and intracellular transport (*Antón et al., 2008*; *Antón et al., 2011*; *Chhabra and Higgs, 2007*; *Gaillard et al., 2011*; *Higgs and Peterson, 2005*; *Ness et al., 2013*; *Pollard, 2007*; *Schönichen and Geyer, 2010*). Phylogenetic analyses indicate that the structure of these proteins is highly conserved (*Figure 1—figure supplement 1*) and examination of evolutionary rates of mammalian formins showed no evidence of positive selection acting on either the INF clade or the INF2 clade (*Table 1*). Several formin family genes have previously been associated with pregnancy and reproductive phenotypes, including preterm birth (*Cruickshank et al., 2013*; *Elovitz et al., 2014*; *Montenegro et al., 2009*). Furthermore,

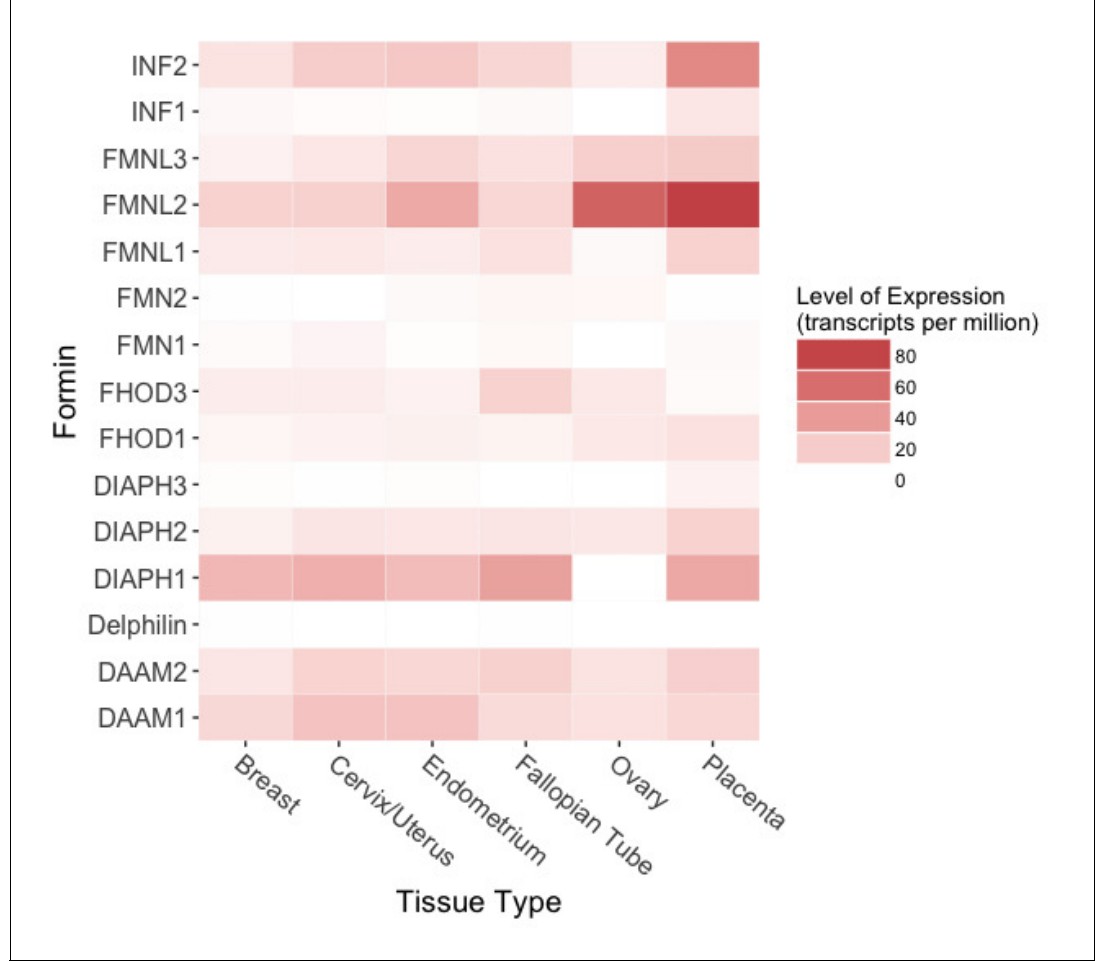

**Figure 1.** Protein expression of formin family members in human female reproductive tissues. Expression levels range from 0.1 to 82.3 transcripts per million (TPM) across the six tissues. Raw data obtained from the Human Protein Atlas database (*Uhlén et al., 2015*).
DOI: https://doi.org/10.7554/eLife.31150.003

The following figure supplement is available for figure 1:

**Figure supplement 1.** Phylogeny of the FH2 domains of 15 formin orthologs across representative primate and model mammal species.
DOI: https://doi.org/10.7554/eLife.31150.004

**Table 1.** Tests of Natural Selection of the INF and INF2 clades.

| Clade | $H_0$ lnL* | $H_1$ lnL[†] | $2\Delta$L[‡] | P value[§] | $\omega$ ratio in background branches in $H_1$ model[#] | $\omega$ ratio in foreground branches in $H_1$ model[¶] |
|---|---|---|---|---|---|---|
| INF | −35554.9 | −3.5554.74 | 0.32 | N.S. | 0.12226 | 0.12995 |
| INF2 | −35554.9 | −35554.56 | 0.68 | N.S. | 0.12422 | 0.11077 |

*Log likelihood score of $H_0$ model, which assumes a single $\omega$ ratio across the phylogeny of the formin family;

[†]Log likelihood score of $H_1$ model, which assumes a single $\omega$ ratio for the foreground clade (INF or INF2) and another $\omega$ ratio for the rest of the branches of the formin phylogeny;

[‡]Difference in log likelihood scores between the $H_0$ and $H_1$ models;

[§]P value of $\chi^2$ test of statistical significance between the the $H_0$ and $H_1$ models;

[#]dn/ds (=$\omega$) ratio of background (all branches except those of the INF or INF2 clade) branches of the formin phylogeny under the $H_1$ model;

[¶]dn/ds (=$\omega$) ratio of foreground (INF or INF2) branches of the formin phylogeny under the $H_1$ model
DOI: https://doi.org/10.7554/eLife.31150.005

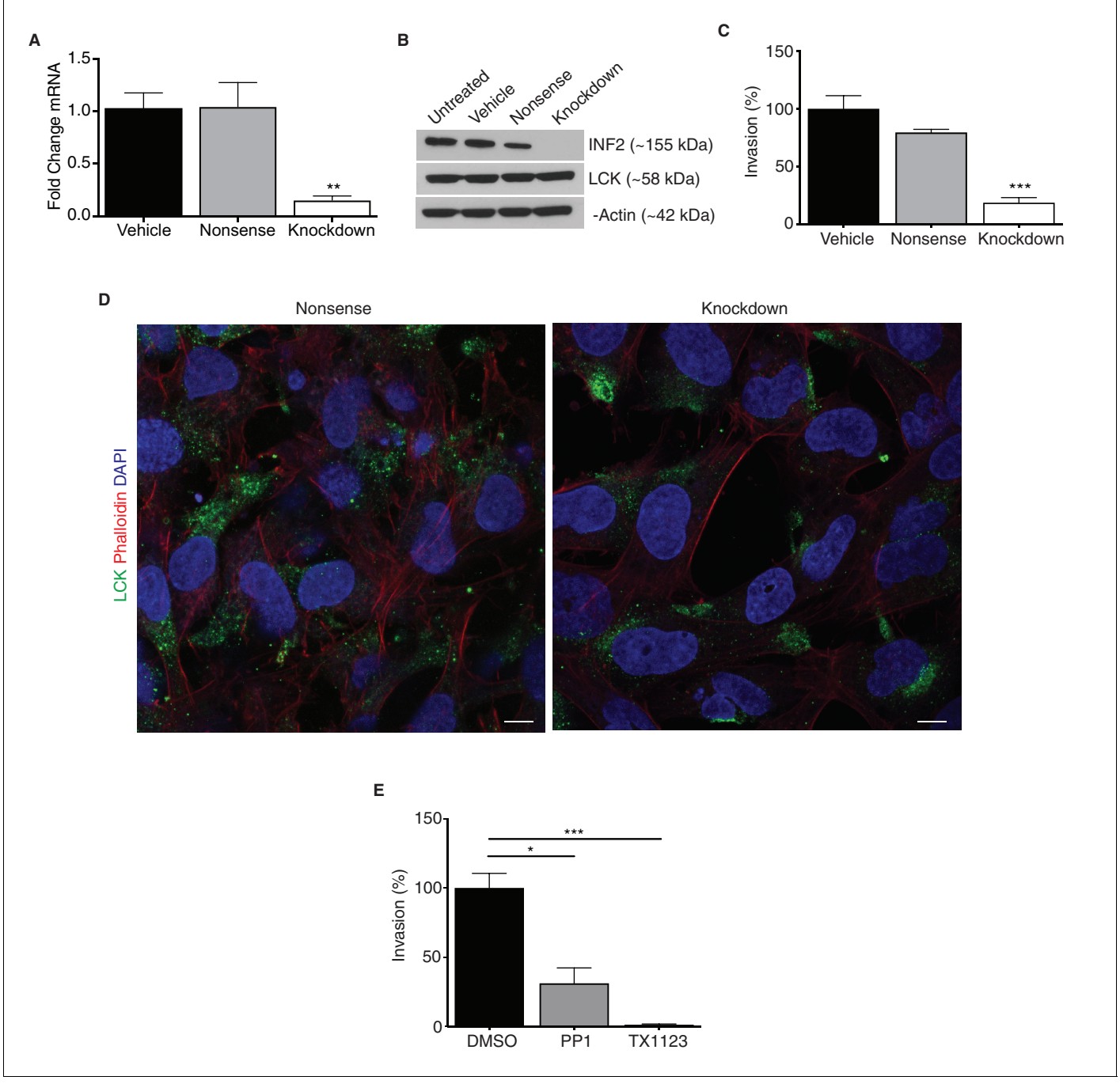

**Figure 2.** *INF2* is necessary for proper EVT invasion and intracellular targeting of LCK. siRNA targeting *INF2* efficiently reduced expression in HTR-8/ SVneo trophoblasts by qPCR (**A**) (n = 4; 1.0 ± 0.14 vs 1.05 ± 0.23 vs 0.15 ± 0.04; **p<0.01) and Western blot analysis (**B**). (**C**) *INF2* reduction in HTR-8/ SVneo cells significantly impeded invasion of these cells through Matrigel (n = 3; 100 ± 11.3 vs 79.44 ± 2.83 vs 18.85 ± 4.46%; ***p<0.001, analyzed by 1-way ANOVA). (**D**) Consistent with results published in Jurkat T lymphocytes, *INF2* reduction restricted LCK to the perinuclear region in cultured EVTs as opposed to cytoplasmic distribution in nonsense siRNA treated EVTs (scale bar: 10 µm). (**E**) Treatment with the LCK/FYN-specific inhibitor PP1 or the SRC inhibitor TX1123 also significantly restricted the ability of these cells to invade (n = 3; 100 ± 10.49 vs 31.23 ± 11.09 vs 1.34 ± 0.51%; *p<0.05, ***p<0.001). All data represent the mean ±SEM and were analyzed by unpaired 2-tailed *t* test, unless otherwise noted.

DOI: https://doi.org/10.7554/eLife.31150.006

The following figure supplements are available for figure 2:

**Figure supplement 1.** Reduction of *INF2* does not alter overall cytoplasmic phalloidin but significantly increases mitochondrial volume in EVTs.

DOI: https://doi.org/10.7554/eLife.31150.007

**Figure supplement 2.** Effect of *INF2* deficiency on MAL2 localization in an in vitro model of human extravillous trophoblast.

*Figure 2 continued on next page*

*Figure 2 continued*

DOI: https://doi.org/10.7554/eLife.31150.008

there is evidence of increased expression of a formin activator, RhoA-GTP, during pregnancy (*Hudson and Bernal, 2012*).

Inverted formin 2 is unique among the formins due to its ability to sever and depolymerize actin filaments in addition to traditional formin functions such as microtubule bundling and actin polymerization (*Chhabra and Higgs, 2006*; *Ramabhadran et al., 2012*). Severing and depolymerization of actin filaments allows INF2 to generate highly transient filaments (*Chhabra et al., 2009*). Transient activation of cofilin—one known regulator of actin filament assembly and disassembly—has been shown to be important in stimulated cell motility (*Yamaguchi and Condeelis, 2007*), suggesting tight regulation of actin dynamics may be vital to extravillous trophoblast invasion. Furthermore, polymerization of actin filaments by INF2 is important for mitochondrial fission, a process that may be important in regulating trophoblast metabolism (*Burton et al., 2017*; *Korobova et al., 2013*). Importantly, INF2 is necessary for intracellular transport—responsible for mobilizing cargo such as SRC kinases, which are responsible for EVT degradation of extracellular matrix (*Patel and Dash, 2012*) and invasiveness. One SRC-like tyrosine kinase trafficked by INF2, lymphocyte-specific protein tyrosine kinase (LCK), has previously been shown to play a role in tumor metastasis (*Andrés-Delgado et al., 2010*; *Mahabeleshwar and Kundu, 2003*)—perhaps one of the many reasons EVT invasion is frequently compared to the metastatic invasion in cancer.

Given the central role of formin proteins in reproduction, actin cytoskeleton dynamics, and the intracellular transport of LCK (*Andrés-Delgado et al., 2010*), we hypothesized that *INF2* is necessary for successful trophoblast invasion and vascular remodeling. *Inf2*-deficient mice demonstrated impaired spiral artery remodeling, hypertension, fetal growth restriction, and altered placental development, identifying the *Inf2* null mouse as a novel model of placental insufficiency.

## Results

### INF2 is necessary for trophoblast invasion through intracellular trafficking of proteins integral for formation of invasive structures

*INF2* targeted siRNA successfully reduced expression in an in vitro model of human EVTs (HTR-8/SVneo; *Figure 2A and B*; p=0.0046). Reduction of *INF2* in these cells did not impact phalloidin content (*Figure 2—figure supplement 1A*; p=0.58), however, mitochondrial volume was significantly increased (*Figure 2—figure supplement 1B and C*; p=0.0048). *INF2* knockdown impaired invasion of HTR-8/SVneo cells by 73% compared to nonsense siRNA- and vehicle-treated cells (*Figure 2C*; p=0.0005). To determine if INF2 is necessary for transcytosis in EVTs as in other cells (*Andrés-Delgado et al., 2010*; *Madrid et al., 2010*), we visualized intracellular localization of MAL2 and LCK in vehicle- or knockdown siRNA-treated HTR-8/SVneo cells. MAL2 is dispersed throughout the cytoplasm in vehicle-treated HTR-8/SVneo cells with no change in localization after INF2 knockdown (*Figure 2—figure supplement 2*). Reduction of *INF2* restricted LCK to the perinuclear region of cultured HTR-8/SVneo cells while LCK was distributed throughout the cytoplasm in controls (*Figure 2D*). There was no change in LCK protein expression (*Figure 2B*). Treatment of HTR-8/SVneo cells with the LCK/FYN-specific inhibitor PP1 reduced invasion by 69% (*Figure 2E*; p=0.011) while the SRC inhibitor TX1123 reduced invasion by 98% compared to controls (*Figure 2E*; p=0.0007).

### Inf2 is temporally regulated in the placenta throughout gestation and is localized to cells of the trophoblast lineage

*Inf2* expression in C57Bl/6J placentas was significantly increased at gestational day 15.5 (E15.5; *Figure 3A*; p=0.026) compared to E13.5 and E18.5 placentas. By E18.5, *Inf2* mRNA returned to earlier pregnancy levels. We demonstrate dense, specific staining of trophoblast cells throughout the labyrinth, junctional zone, and decidua in control mice and none in the knockout mice (*Figure 3B*). Lck is restricted to the perinuclear region in *Inf2*$^{-/-}$ trophoblasts while it is distributed throughout the cytoplasm in *Inf2*$^{+/+}$ trophoblasts (*Figure 3C*). Immunofluorescence staining revealed co-

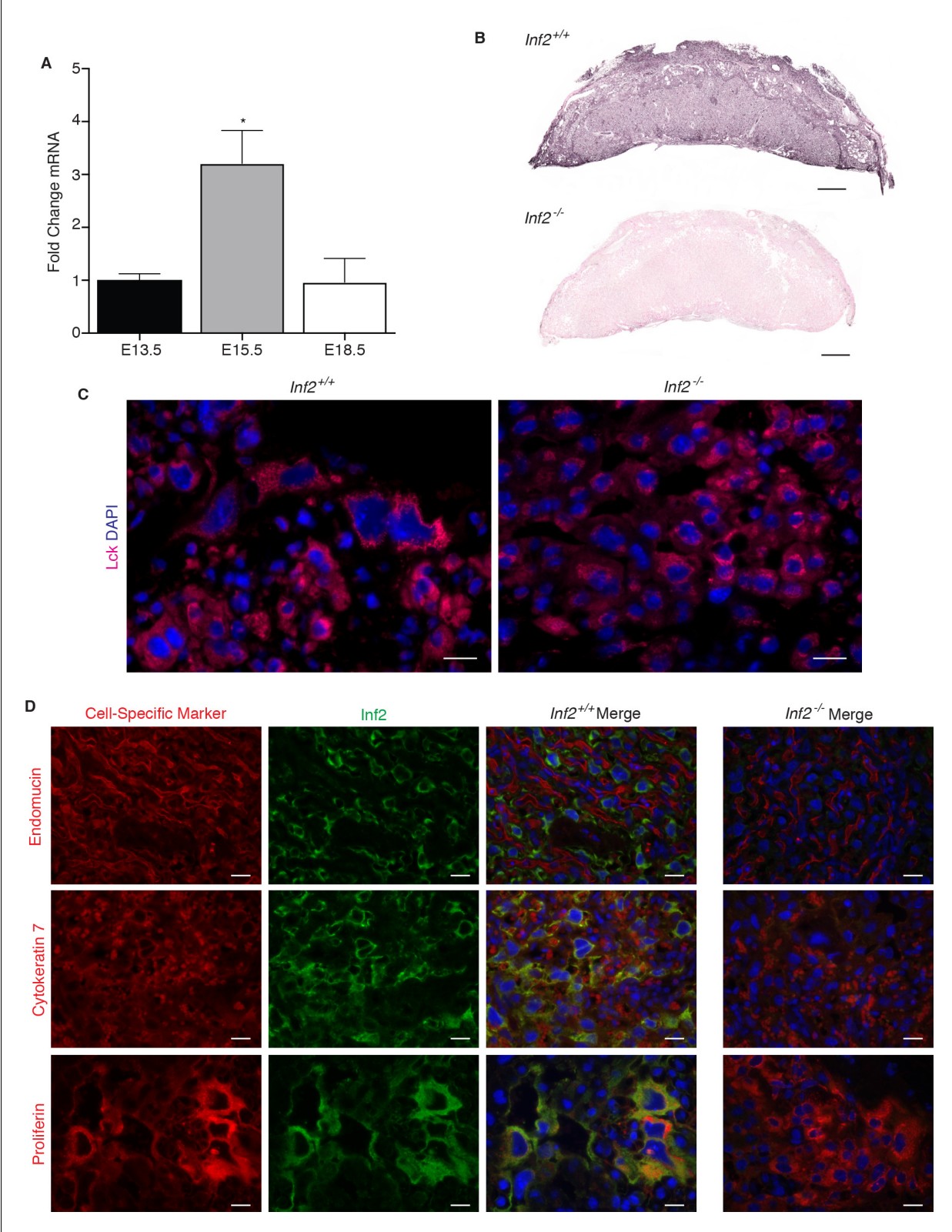

**Figure 3.** *Inf2* is highly expressed in the mouse placenta and co-localizes with trophoblast markers. (**A**) Timecourse of *Inf2* mRNA levels in C57Bl/6J mice at E13.5, E15.5, and E18.5 (n = 2, 5, 5; 1.01 ± 0.12 vs 3.21 ± 0.62 vs 0.96 ± 0.45; *p<0.05 by 1-way ANOVA). (**B**) IHC of E15.5 placentas reveal dense, specific staining of Inf2 throughout the *Inf2+/+* labyrinth, junctional zone, and decidua with no positive staining in the *Inf2−/−* placenta (scale bar: 500 μm). (**C**) Consistent with our in vitro data, at E15.5, Lck is localized throughout trophoblast cells in the *Inf2+/+* placenta while it is mostly perinuclear in

*Figure 3 continued on next page*

*Figure 3 continued*

*Inf2⁻/⁻* placentas (scale bar: 50 μm). (D) Inf2 does not co-localize with endothelial cell marker endomucin, but co-localizes with the pan-trophoblast marker cytokeratin-7 and the TGC marker proliferin in *Inf2⁺/⁺* E15.5 placentas (scale bar: 50 μm). All data represent the mean ±SEM.

DOI: https://doi.org/10.7554/eLife.31150.009

localization of Inf2 with the pan-trophoblast marker cytokeratin-7 (Ck7) and the trophoblast giant cell (TGC) marker proliferin (*Figure 3D*).

## Improper spiral artery remodeling in Inf2⁻/⁻ placentas causes systemic hypertension during pregnancy that resolves after delivery

We visualized lectin-labeled maternal spiral arteries in cleared, depth-coated placentas. Fully extended spiral artery numbers were counted in placentas rendered in 3D (*Videos 1* and *2*). At E19.0, the number of spiral arteries in *Inf2⁻/⁻* placentas was significantly reduced compared to wild-type placentas (*Figure 4A and B*; p=0.023). Using the volumetric pressure cuff system to monitor blood pressure changes throughout pregnancy, systolic blood pressure dropped from pre-pregnancy levels at E15.5 in all females. In contrast, at E17.5 blood pressure was significantly elevated in *Inf2⁻/⁻* females compared to *Inf2⁺/⁺* (*Figure 4C*; p=0.012). By postnatal day 2 (P2), the systolic blood pressure of all females was comparable to pre-pregnancy levels. No significant differences in total urinary protein were measured in non-gravid females (n = 6; 26984 ± 2936 vs 29428 ± 3441 ng/μL) or females at E17.5 (n = 3, 4; 33834 ± 4644 vs 34727 ± 7028 ng/μL; data not shown).

As maternal hypertension in pregnancy may result from abnormal placental production of angiogenic factors, we measured these levels in serum. Despite a trend of higher placental growth factor-2 (Plgf-2) in the maternal circulation of *Inf2⁻/⁻* females at E15.5 (*Figure 4—figure supplement 1*; p=0.16), no significant differences were detected in either Plgf-2 (p=0.16, 0.97) or FMS-like tyrosine kinase 1 (Flt1; p=0.90, 0.87) levels at E15.5 or E18.5.

## Inf2 is vital for the regulation of gestation length and fetal growth

To evaluate the significance of *Inf2* in gestation, we compared pregnancy outcomes in *Inf2⁺/⁺* and *Inf2⁻/⁻* mice. Gestation length was increased by 9.8 hr in *Inf2⁻/⁻* mice (*Figure 5A*; p=0.009) with no impact on pup weight at birth (p=0.96), litter size (p=0.83), or total litter weight (*Figure 5B* and *Figure 5—figure supplement 1A and B*; p=0.51 at E18.5, 0.31 at p0). Despite extended gestational length, there were no detectable differences in serum progesterone (p=0.64), uterine prostaglandins F2α (p=0.64) and E2 (p=0.99), or oxytocin receptor mRNA expression at E18.5 (*Figure 5—figure supplement 2A–D*; p=0.33) (*Bezold et al., 2013*). While normal weight at birth, fetal weight at E18.5 was significantly reduced in *Inf2⁻/⁻* pups compared to *Inf2⁺/⁺* pups (*Figure 5B*; p=0.020). There were no differences in placental weight (*Figure 5C*; p=0.82); however, the ratio of fetal to placental weight was significantly reduced in *Inf2⁻/⁻* mice (*Figure 5D*; p=0.019). Previous studies showed that altered fetal growth in late pregnancy is preceded by changes in placental

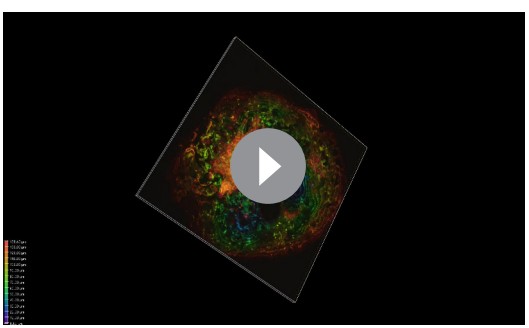

**Video 1.** *Inf2⁺/⁺* placenta at E19.0.
DOI: https://doi.org/10.7554/eLife.31150.012

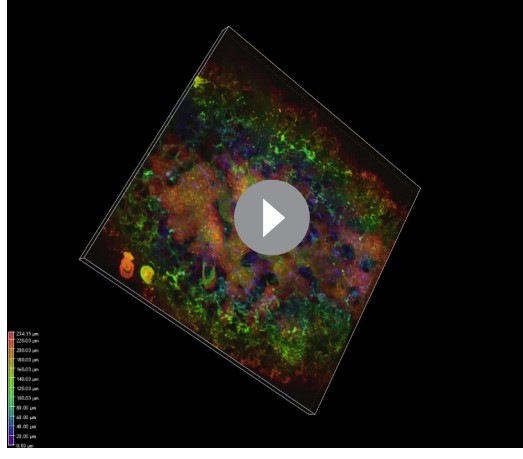

**Video 2.** *Inf2⁻/⁻* placenta at E19.0.
DOI: https://doi.org/10.7554/eLife.31150.013

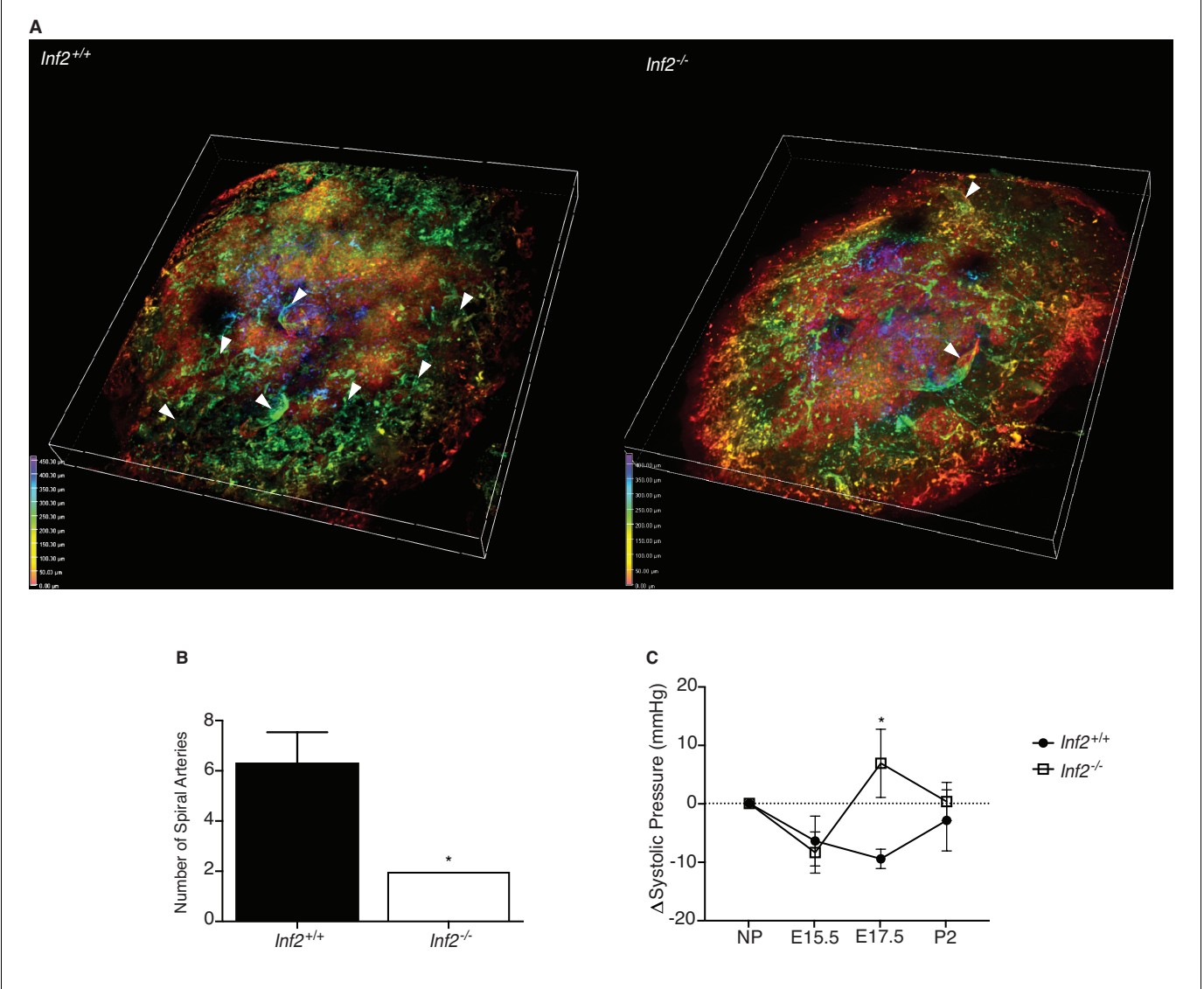

**Figure 4.** Loss of *Inf2* alters maternal spiral artery remodeling, resulting in systemic hypertension late in pregnancy. (**A**) Lectin-tagged maternal spiral arteries (arrowheads) were visualized in E19.0 placentas after clearing. Positive staining was depth coded in the 3D image based on position in Z (0.00 μm in red, 50.00 μm in orange, 150.00 μm in yellow, 250.00 μm in green, 350.00 μm in cyan, 400.00 μm in indigo, and 450.00 μm in violet). (**B**) The number of fully extended spiral arteries was quantified and found to be significantly reduced in *Inf2*$^{-/-}$ placentas compared to wildtype placentas (n = 3; 6.33 ± 1.2 vs 2 ± 0; *p<0.05). (**C**) Calculated as change (Δ) from the non-pregnant state (NP; n = 9, 8; 0.00 ± 0.00 mmHg), the systolic blood pressure of both *Inf2*$^{+/+}$ and *Inf2*$^{-/-}$ females decreased at E15.5 (-6.422 ± 4.262 vs −8.395 ± 3.523 mmHg). At E17.5, blood pressure was significantly elevated in *Inf2*$^{-/-}$ females, while *Inf2*$^{+/+}$ blood pressure remained unchanged (−9.468 ± 1.650 vs 6.871 ± 5.834 mmHg; *p<0.05). By P2, both *Inf2*$^{+/+}$ and *Inf2*$^{-/-}$ systolic blood pressure returned to pre-pregnancy levels (−2.902 ± 5.222 vs 0.331 ± 3.266 mmHg). All data represent the mean ±SEM and were analyzed by unpaired 2-tailed *t* test, unless otherwise noted.

DOI: https://doi.org/10.7554/eLife.31150.010

The following figure supplement is available for figure 4:

**Figure supplement 1.** Effect of *Inf2* deficiency on angiogenic factors in serum.
DOI: https://doi.org/10.7554/eLife.31150.011

---

nutrient transport (*Jansson et al., 2006*), however, there were no detectable differences in mRNA expression of the amino acid or glucose transporters studied here at E18.5 (*Figure 5—figure supplement 3A–D*; p=0.19–0.65).

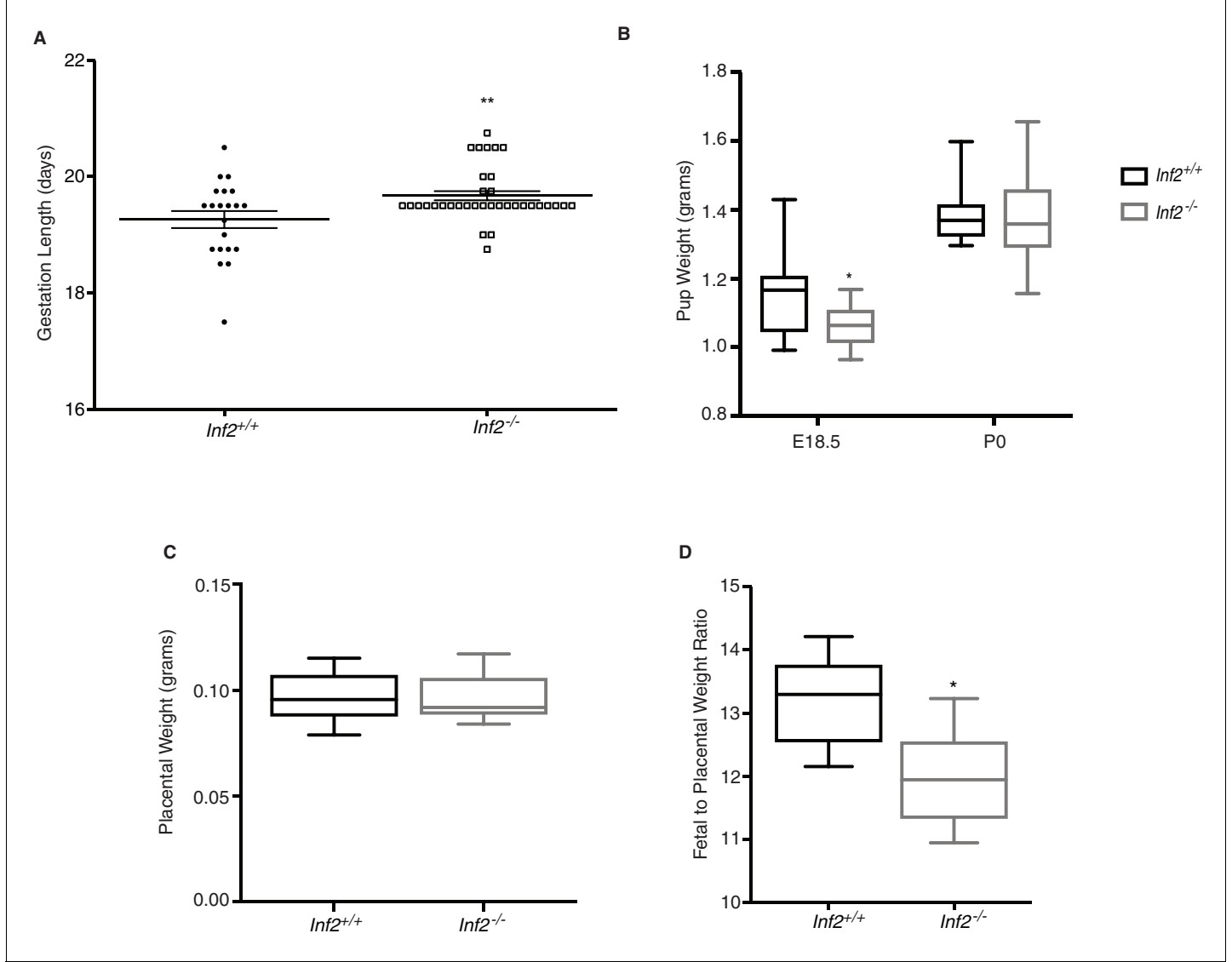

**Figure 5.** Murine *Inf2* is important for regulating gestation length and fetal growth. (**A**) Gestation lengths measured from visualization of a copulatory plug (n = 21, 35 dams; 19.26 ± 0.15 vs 19.67 ± 0.08 days; **p<0.01). (**B**) Pup weights at E18.5 were significantly reduced in *Inf2*$^{-/-}$ dams (n = 15, 13; 1.152 ± 0.03 vs 1.062 ± 0.018 grams; *p<0.05) while no difference in pup weight was measured at time of birth (P0; n = 10, 32; 1.358 ± 0.013 vs 1.359 ± 0.01 grams). (**C**) No significant differences in placental weight at E18.5 were detected (n = 13; 0.096 ± 0.003 vs 0.096 ± 0.003 grams). (**D**) The ratio of fetal weight to placental weight was significantly reduced in *Inf2*$^{-/-}$ dams (n = 6; 13.18 ± 0.298 vs 11.947 ± 0.326; *p<0.05). Data are presented as a boxplot (median, interquartile range, minimum, and maximum). All data represent the mean ±SEM and were analyzed by unpaired 2-tailed *t* test, unless otherwise noted.

DOI: https://doi.org/10.7554/eLife.31150.014

The following figure supplements are available for figure 5:

**Figure supplement 1.** Effects of loss of *Inf2* on number of pups from live births and total litter weight.
DOI: https://doi.org/10.7554/eLife.31150.015
**Figure supplement 2.** Effect of *Inf2* deficiency on systemic and local indicators of labor.
DOI: https://doi.org/10.7554/eLife.31150.016
**Figure supplement 3.** Effect of *Inf2* deficiency on placental nutrient transporter mRNA.
DOI: https://doi.org/10.7554/eLife.31150.017

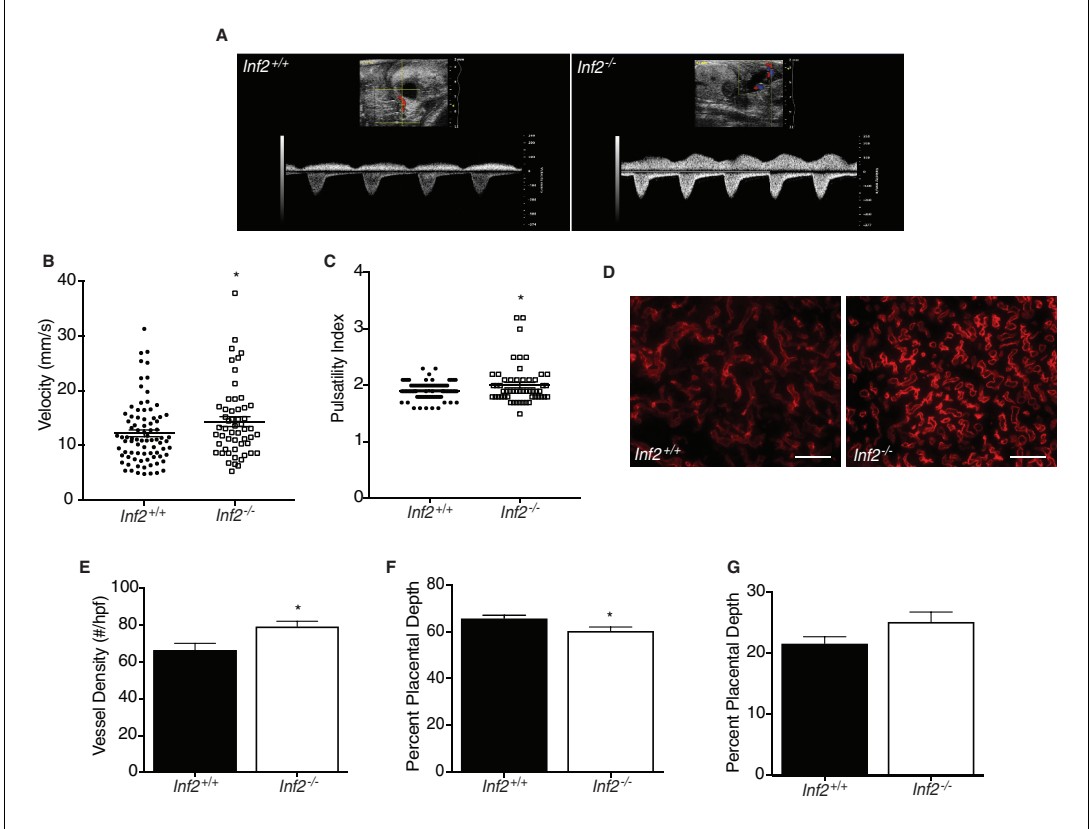

**Figure 6.** Loss of *Inf2* alters placental vascularization, impeding function and umbilical blood flow. (**A**) Umbilical Doppler images from *Inf2⁺/⁺* and *Inf2⁻/⁻* fetuses highlighting differences in arterial and venous waveforms at E18.5. End diastolic velocity (**B**) (12.03 ± 0.619 vs 14.15 ± 0.902 mm/s) and pulsatility index (**C**) (1.907 ± 0.016 vs 2.007 ± 0.047) are significantly increased in *Inf2⁻/⁻* fetuses (n = 83, 54 fetuses; *p<0.05). (**D**) Representative images from endomucin-labeled *Inf2⁺/⁺* and *Inf2⁻/⁻* E18.5 placentas depict differences in vessel density (scale bar: 50 μm), quantified in (**E**) (n = 2–3 placentas per dam, 5 and 7 dams; 66.53 ± 3.65 vs 79.4 ± 2.83 number/high powered field; *p<0.05). (**F**) The percent of total placenta depth consisting of the labyrinth was significantly reduced at E18.5 (n = 3 placentas per dam, five dams per genotype; 65.72 ± 1.26 vs 60.38 ± 1.58%; *p<0.05) while no differences were measured in the junctional zone (**G**) (n = 3 placentas per dam, five dams per genotype; 21.6 ± 1.09 vs 25.15 ± 1.60%). All data represent the mean ±SEM and were analyzed by unpaired 2-tailed *t* test, unless otherwise noted.

DOI: https://doi.org/10.7554/eLife.31150.018

The following figure supplement is available for figure 6:

**Figure supplement 1.** Results of *Inf2* loss on fetal health.
DOI: https://doi.org/10.7554/eLife.31150.019

## Inf2⁻/⁻ pregnancies are complicated by placental vasculopathy

Altered end-diastolic flow and pulsatility index may indicate the presence of intrauterine growth restriction (IUGR) and/or PE (*Bond et al., 2015*; *Krebs et al., 1996*; *Turan et al., 2008*). To assess vascular capacity and placental function, we performed umbilical artery and vein Doppler in pregnant *Inf2⁺/⁺* and *Inf2⁻/⁻* dams at E18.5 (*Figure 6A*). End-diastolic velocity (EDV) and pulsatility index (PI) were significantly elevated in *Inf2⁻/⁻* fetuses (*Figure 6B and C*; p=0.045 and 0.022), with no significant differences in resistance index (p=0.33), peak systolic velocity (p=0.12), or fetal heart rate (*Figure 6—figure supplement 1A–C*; p=0.06). Moreover, some umbilical vein waveforms appeared pulsatile (*Figure 6A*). Fetal vascular density in the labyrinth of placentas (*Figure 6D*) at E18.5 was significantly higher in *Inf2⁻/⁻* placentas (*Figure 6E*; p=0.018) and the proportion of placental depth consisting of labyrinth but not junctional zone was significantly reduced in *Inf2⁻/⁻* compared to *Inf2⁺/⁺* placentas (*Figure 6F and G*; p=0.030 and 0.105).

To determine if *INF2* regulates angiogenic factor expression, we utilized an in vitro model of the crosstalk between CTBs (BeWo choriocarcinoma) with reduced *INF2* mRNA expression (*Figure 7A*; p<0.0001) and human placental microvascular endothelial cells (HPMVECs). Knockdown of *INF2*

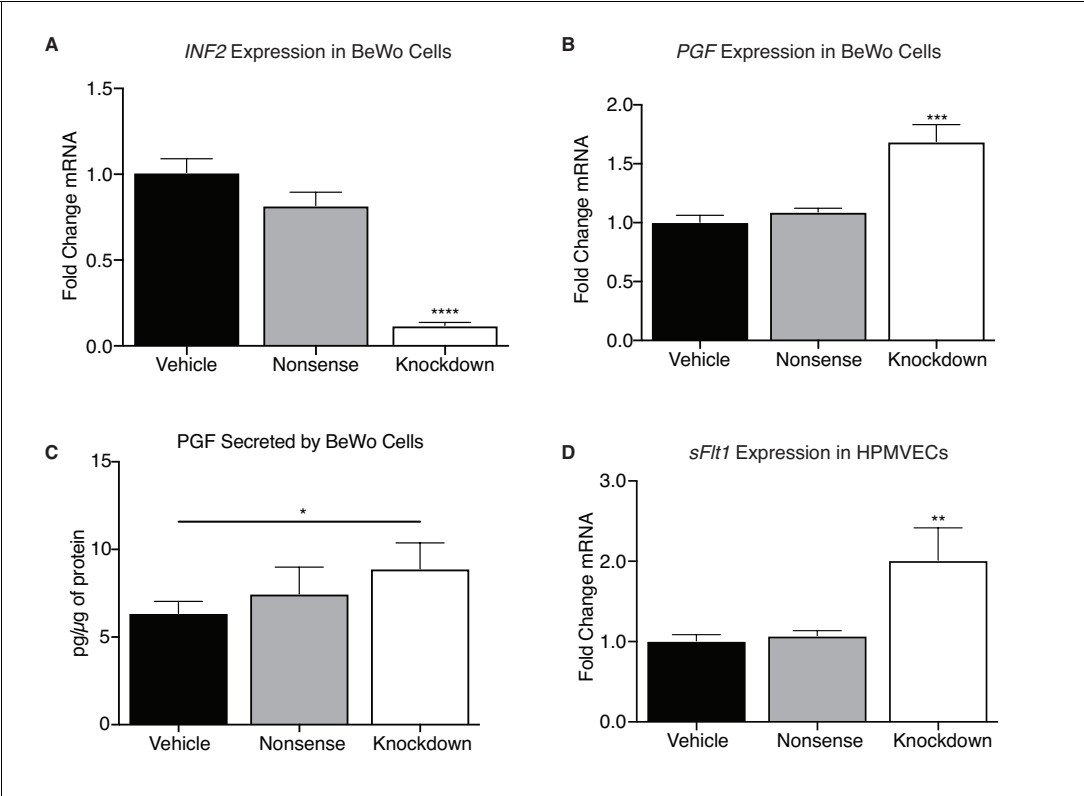

Figure 7. *INF2* is necessary for regulating angiogenic factor expression. (A) Knockdown of *INF2* in the BeWo cells (n = 4, 5, 5; 1.0 ± 0.08 vs 0.82 ± 0.08 vs 0.12 ± 0.02; ****p<0.0001, analyzed by 1-way ANOVA) significantly increased *PGF* mRNA (B) (n = 4, 5, 5; 1.0 ± 0.06 vs 1.09 ± 0.03 vs 1.69 ± 0.14; ***p<0.001, analyzed by 1-way ANOVA). This increase in mRNA corresponded with an increase in secreted PGF by *INF2*-deficient BeWo cells compared to vehicle-treated cells (D) (n = 7; 6.353 ± 0.68 vs 7.462 ± 1.53 vs 8.888 ± 1.485 pg/μg protein; *p<0.05, analyzed by paired 1-tailed *t* test). Treatment with nonsense siRNA did not significantly alter secretion of PGF compared to vehicle. Conditioned media from *INF2*-deficient BeWo cells induced a significant increase in *sFLT1* mRNA (D) (n = 4, 5, 3; 1.0 ± 0.15 vs 1.07 ± 0.06 vs 2.01 ± 0.40; **p<0.01, analyzed by 1-way ANOVA). All data represent the mean ±SEM and were analyzed by unpaired 2-tailed *t* test, unless otherwise noted.
DOI: https://doi.org/10.7554/eLife.31150.020

significantly increased *PGF* mRNA in the BeWo cell line (*Figure 7B*; p=0.0007). HPMVEC exposure to cultured media significantly increased *soluble vascular endothelial growth factor receptor type 1* (*sVEGFR-1; sFLT1*) mRNA (*Figure 7D*; p=0.0083) in response to *INF2* deficiency; therefore, we hypothesized that PGF protein secreted by *INF2*-knockdown BeWo cells would also be increased and underlie the *sFLT1* response in the HPMVECs. Knockdown of *INF2* significantly upregulated secretion of PGF compared to vehicle-treated cells (*Figure 7C*; p=0.037). PGF secretion by nonsense siRNA-treated cells, however, did not differ significantly from vehicle-treated cells (*Figure 7C*; p=0.39). Global loss of *Inf2* in mice, however, did not change placental *Pgf* or *sFlt1* mRNA levels in vivo (data not shown).

## Discussion

Establishment of a healthy pregnancy is dependent on proper embryo implantation and the differentiation, invasion, and communication of trophoblast cells with the uterine milieu—processes that continue throughout gestation as the placenta develops and grows (*Kokkinos et al., 2010*).

Alteration in CTB differentiation disrupts placental architecture (*Cross, 2005*) and changes in placental vascularization results in placental insufficiency. At E18.5, *Inf2*-deficient fetuses were significantly growth restricted and, interestingly, the ratio of fetal weight to placental weight was significantly reduced in these mice—consistent with IUGR in human pregnancies as a result of inefficient placentas (*Hayward et al., 2016*). Unlike the human situation, birth weights of growth-restricted *Inf2*⁻/⁻ pups did not differ from wildtype pups, which we hypothesize is due to the

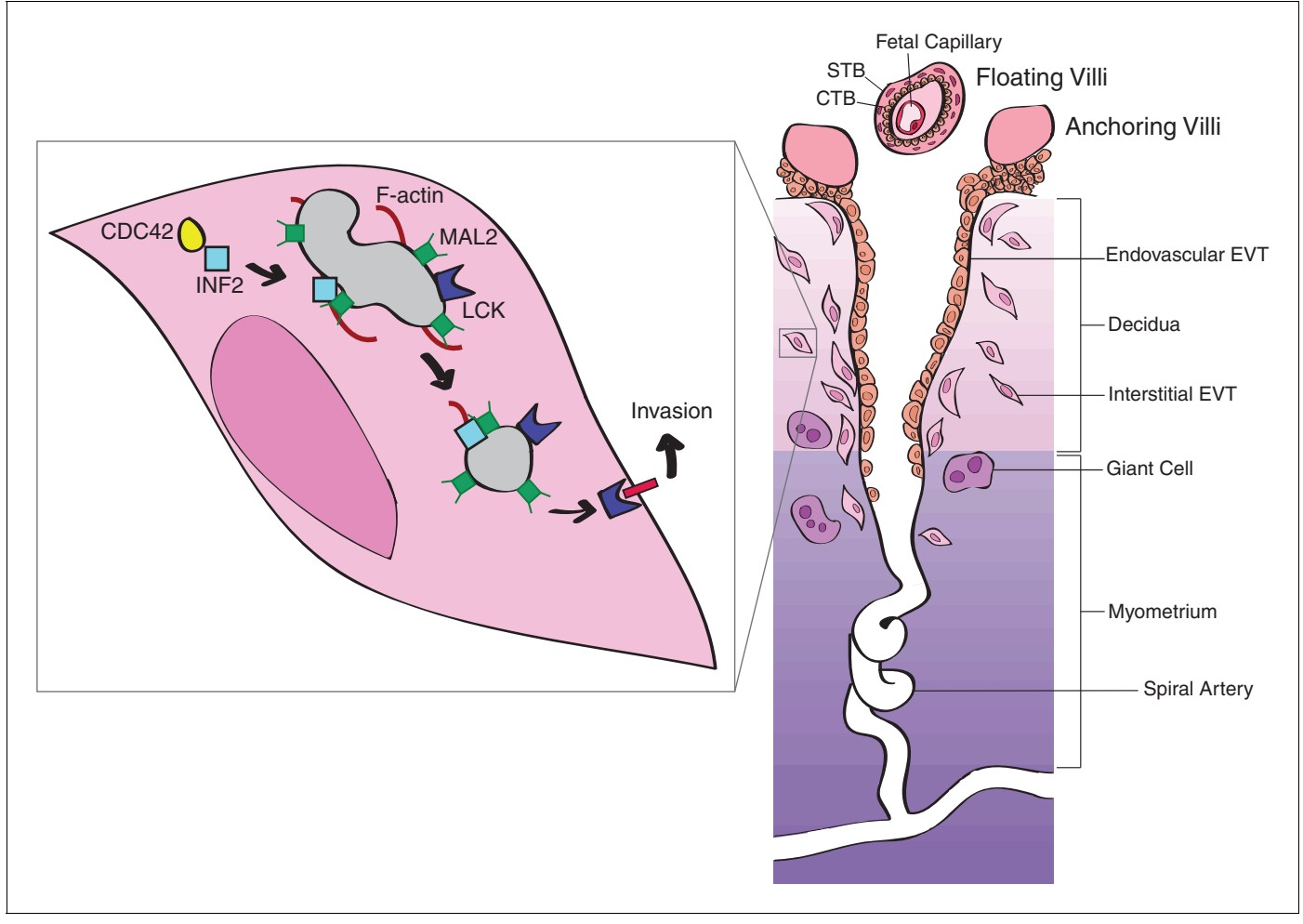

**Figure 8.** Proposed model of INF2-mediated trophoblast invasion and spiral artery remodeling. Intracellular transport along microtubule tracks is facilitated by the binding of INF2 to MAL2-coated vesicles or lipid rafts. By binding microtubules and active CDC42, INF2 regulates formation of actin filaments, driving transport of vesicles (*Antón et al., 2008*; *Antón et al., 2011*; *Ness et al., 2013*). LCK cargo is transported to the plasma membrane, causing cytoskeletal changes necessary for EVT invasion and, consequently, spiral artery remodeling (*Moffett-King, 2002*). In the absence of INF2, LCK is restricted to the perinuclear region of the trophoblast, preventing activation of the signaling cascade necessary for formation of invasive actin-rich structures. Failure of invasion impedes spiral artery remodeling, leading to disease. Figure based on (*Moffett-King, 2002*). Reprinted with permission from Macmillan Publishers Ltd: Nature Reviews Immunology (*Moffett-King, 2002*), copyright 2002.

DOI: https://doi.org/10.7554/eLife.31150.021

increased gestation length in $Inf2^{-/-}$ animals allowing these growth restricted fetuses to catch up in weight prior to birth. We did not detect change in placental nutrient transporter mRNA expression at E18.5, however previous studies have demonstrated that alterations in placental nutrient transporters may occur prior to changes in birth weight and not concurrently (*Jansson et al., 2006*). Therefore, it is possible that $Inf2$ may play a role in nutrient transporter localization and should be interrogated in future studies.

Umbilical ultrasound at E18.5 revealed significant increases in end-diastolic velocity and pulsatility index in $Inf2^{-/-}$ fetuses, consistent with IUGR and poor placental function. A reduction in labyrinth depth in $Inf2^{-/-}$ placentas with a significant increase in fetal vessel density confirms aberrant placental architecture and placental vasculopathy. Placental vascularization and development depends on both autocrine and paracrine signaling between trophoblasts and endothelial cells (*Charnock-Jones and Burton, 2000*; *Ong et al., 2000*; *Troja et al., 2014*). Reduction of *INF2* in BeWo cells is sufficient both increase endogenous *PGF* expression and secretion, indirectly increasing endothelial *sFLT1* mRNA expression. These data suggest *INF2* is necessary for angiogenic balance in trophoblasts, however, further studies to determine the precise mechanisms by which INF2 modulates

expression and secretion of these factors in trophoblasts are required. While global loss of *Inf2* in vivo did not reflect this, we posit that cell-specific differences that are easily ascertained in cell culture are masked by the in vivo milieu.

Differentiation and invasion of EVTs is critical for normal spiral artery remodeling and placentation (*Moffett-King, 2002*). Shallow invasion by EVTs is thought to underlie failed remodeling, the primary placental insult leading to ischemia and subsequent development of PE and maternal hypertension. We report that loss of *Inf2* significantly reduced the number of fully extended maternal spiral arteries, suggesting shallow invasion and remodeling by TGCs. Like other models of PE, *Inf2*$^{-/-}$ mice display new onset of maternal hypertension in late pregnancy that resolves after delivery—consistent with the peripheral vasoconstriction and reduced arterial compliance seen in patients with PE. However, we did not detect any differences in total urinary protein or serum pro- and anti-angiogenic factors in mid- or late gestation in *Inf2*$^{-/-}$ mice as reported in patients clinically (*Tsatsaris et al., 2003*). Proteinuria may be masked in our transgenic line, however, as C57BL/6 mice are resistant to developing proteinuria (*Ishola et al., 2006*). Women with PE and severe IUGR commonly deliver preterm due to clinical intervention to avoid maternal and fetal demise; therefore, it is unknown whether the length of gestation would naturally be increased in these women as it is in our mouse model to allow for continued growth, resulting in an average weight at birth. These data suggest a limitation—that despite shallow invasion, limited spiral artery remodeling, and hypertension in late gestation, our model may not fully recapitulate human PE but may represent a model of placental insufficiency with maternal hypertension.

Successful tumor cell invasion is dependent on the generation of invasive actin-rich structures (*Parast et al., 2001*; *Patel and Dash, 2012*), such as invadopodia. Formation of invadopodia involves cytoskeletal remodeling, suggesting that this process may also be essential for EVT invasion. INF2 is a cytoskeletal modulator that is important for regulating cell polarity, mitochondrial fission, intracellular trafficking, as well as cell and tissue morphogenesis (*Chhabra and Higgs, 2006*; *Chhabra et al., 2009*; *Goode and Eck, 2007*; *Madrid et al., 2010*). Despite its major function in regulating cellular actin dynamics, reduction of *INF2* did not alter overall cytoskeletal F-actin as determined by comparing cytoplasmic phalloidin content in EVTs. In functional assays, this reduction significantly altered invasion, suggesting EVT invasion is dependent on *INF2* expression but independent of cytoskeletal F-actin maintenance. Pharmacologic inhibition of the SRC-like tyrosine kinase LCK—which is dependent on INF2 for proper intracellular trafficking (*Andrés-Delgado et al., 2010*)—similarly reduced invasion. Consistent with published results in human T lymphocytes (*Andrés-Delgado et al., 2010*), reduction of *INF2* in HTR-8/SVneo cells restricted LCK to the perinuclear region, which was reflected in *Inf2*$^{-/-}$ placentas. Together, these data suggest a novel model by which EVT invasion is mediated by INF2-dependent targeting of LCK to the plasma membrane (*Figure 8*). Loss of *Inf2* and the subsequent restriction of Lck in the cytoplasm would inhibit TGC invasion and spiral artery remodeling in the mouse, leading to maternal hypertension late in gestation.

Consistent with previous studies, loss of *INF2* significantly increased mitochondrial size in vitro (*Korobova et al., 2013*). Our data, therefore, supports a role for INF2 in mitochondrial fission. Studies suppressing dynamin-related protein 1 (Drp1)—a fission protein whose recruitment to mitochondria is facilitated by INF2 (*Jin et al., 2017*; *Korobova et al., 2013*)—in breast cancer cells inhibited formation of lamellipodia and suppressed their migration and invasion capabilities (*Zhao et al., 2013*). However, as knockdown of Drp1 results in a significantly more severe mitochondrial phenotype, we support the suggestion put forth by Korobova et al. that there are INF2-independent pathways in mitochondrial fission. Therefore, we surmise that INF2-dependent trophoblast invasion seen in our study is not only mitochondrial-dependent. This presents an alternative mechanism in the regulation of invasion that requires meticulous dissection between mitochondrial dynamics and metabolism, the precise role of INF2 in mitochondrial fission, and mitochondrial recruitment to lamellipodia in EVTs.

Overall, our data represent one mechanism of EVT invasion and proffer new avenues for future study in the molecular biology of trophoblast cells of the placenta. We present a novel genetic model of placental insufficiency encompassing critical aspects of IUGR and late-onset PE, despite several differences between the mouse and human placenta.

# Materials and methods

## Key resources table

| Reagent type | Designation | Source or reference | Identifiers |
|---|---|---|---|
| Gene (*H. sapiens*) | *INF2* | N/A | N/A |
| Gene (*M. musculus*) | *Inf2* | N/A | N/A |
| Strain | C57BL/6 | Jackson Laboratories | RRID:IMSR_JAX:000664 |
| | *Inf2⁻/⁻*; *Inf2* KO | KOMP | RRID:MGI:5759294 |
| Genetic reagent (*H. sapiens*) | Nonsense siRNA | Millipore Sigma SIC001 | N/A |
| | Knockdown siRNA | ThermoFisher Scientific 4392420 | N/A |
| Cell line (*H. sapiens*) | HTR-8/SVneo | Charles Graham; ATCC | RRID:CVCL_7162 |
| | BeWo | ATCC | RRID:CVCL_0044 |
| | HPMVEC | Helen Jones | N/A |
| Antibody | Rabbit polyclonal anti-MAL2 | Abcam | RRID:AB_1280985 |
| | Rabbit polyclonal anti-Lck | Abcam ab208787 | N/A |
| | Rabbit polyclonal anti-INF2 | MilliporeSigma | RRID:AB_1078325 |
| | Goat polyclonal anti-Endomucin | R&D Systems | RRID:AB_2100035 |
| | Gloat polyclonal anti-Proliferin | R&D Systems | RRID:AB_2284428 |
| | Rabbit polyclonal anti-LCK | Abcam | RRID:AB_2249950 |
| | MitoTracker Red CMXRos | ThermoFisher Scientific M7512 | N/A |
| | Rabbit polyconal anti-INF2 | MilliporeSigma | RRID:AB_11203139 |
| Primers | Mouse *Inf2* qPCR primers | Forward: CGAGTAGTTGACCACCGAGG<br>Reverse: ACAGCACTCTGCACCATCTC | N/A |
| | Mouse *Rps20* qPCR primers | Forward: GCTGGAGAAGGTTTGTGCG<br>Reverse:AGTGATTCTCAAAGTCTTGGTAGGC | N/A |
| | Mouse *Oxtr* qPCR primers | Forward: ACGGGTCAGTAGTGTCAAGC<br>Reverse: TAATGCTCGTCTCTCCAGGC | N/A |
| | Mouse *Slc38a2* qPCR primers | Forward: ACCTTTGGTGATCAAGGCAT<br>Reverse: AGGACCAGATAGTCACCGTT | N/A |
| | Mouse *Slc2a1* qPCR primers | Forward: TGCAGTTCGGCTATAACACT<br>Reverse: GTAGCGGTGGTTCCATGTTT | N/A |
| | Mouse *Slc2a3* qPCR primers | Forward: CTTTGGCAGACGCAACTCTA<br>Reverse: GCTATCTTGGCGAATCCCAT | N/A |
| | Mouse *Slc2a4* qPCR primers | Forward: ACTGGACCTGTAACTTCAT<br>Reverse: GCAAATAGAAGGAAGACGTA | N/A |
| | Mouse *Pgf* qPCR primers | Forward: GACCTATTCTGGAGACGACA<br>Reverse: GGTTCCTCAGTCTGTGAGTT | N/A |
| | Mouse *sFlt1* qPCR primers | Forward: TGACGGTCATAGAAGGAACA<br>Reverse: TAGTTGGGGATAGGGAGCCA | N/A |
| | Human *INF2* qPCR primers | Forward: CACATCCAACGTGATGGTGAAG<br>Reverse: GGAGAGCTCGTTCATGACAATG | N/A |
| | Human *ACTB* qPCR primers | Forward: CGCGAGAAGATGAACCAG<br>Reverse: TAGCACAGCCTGGATAGCAA | N/A |
| | Human *PGF* qPCR primers | Forward: GAGGAGAGAGAAGCAGAGA<br>Reverse: GTGACGGTAATAAATACACGAG | N/A |
| | Human *sFLT1* qPCR primers | Forward: AGAAGGGCTCTGTGGAAAGT<br>Reverse: ACACAGGTGCATGTTAGAGTG | N/A |

*Continued on next page*

*Continued*

| Reagent type | Designation | Source or reference | Identifiers |
|---|---|---|---|
| Commercial assay or kit | Mouse Angiogenesis/Growth Factor Magnetic Bead Panel | MilliporeSigma MAGPMAG-24K | N/A |
| | Mouse Soluble Cytokine Receptor Magnetic Bead Panel | MilliporeSigma MSCRMAG-42K | N/A |
| | Progesterone Mouse/Rat ELISA | BioVendor RTC008R | N/A |
| | PGE2 EIA Kit | Oxford Biomedical Research EA02 | N/A |
| | PGF2a EIA Kit | Oxford Biomedical Research EA05 | N/A |
| | Human PLGF Quantikine ELISA Kit | R and D Systems DPG00 | N/A |
| Chemical compound, drug | PP1 | Cayman Chemical 14244 CAS: 172889-26-8 | N/A |
| | TX1123 | MilliporeSigma 655200 CAS: 157397-06-3 | N/A |
| Software, algorithm | MAFFT, v7.310 | (*Katoh and Standley, 2013*) | RRID:SCR_011811 |
| | SeaView | (*Gouy et al., 2010*) | RRID:SCR_015059 |
| | RAxML, v8.2.9 | (*Stamatakis, 2014*) | RRID:SCR_006086 |
| | PROTGAMMAAUTO | (*Jones et al., 1992*) | N/A |
| | FigTree, v1.4.3 | (*Rambaut, 2007*) | RRID:SCR_008515 |

## Formin expression in human tissues

Expression data on the 15 human formins in placenta, fallopian tube, breast, ovary, endometrium, and uterus were collected from the Human Protein Atlas (*Uhlén et al., 2015*) where RNA-sequence expression is measured in TPM (Transcripts Per Kilobase Million).

## Sequence retrieval and phylogenetic inference

To identify genes that belong to the formin gene family we used the defining FH2 domain as a query against the Pfam database (*Finn et al., 2005*). Using UniProtKB identifiers for 11 mammalian species (*Homo sapiens, Pan troglodytes, Gorilla gorilla, Pongo abelii, Nomascus leucogenys, Macaca mulatta, Calithrix jacchus, Otolemur garnettii, Mus musculus, Rattus norvegicus,* and *Canis lupus familiaris*), coding and protein sequences of the 15 genes of the formin family were collected from ENSEMBL, release 89 (*Cunningham et al., 2015*). Exceptions to this gene family were found in *P. troglodytes* and *P. abelii* (*P. troglodytes* lacks DIAPH1 and FMN2 and *P. abelii* lacks FMNL1). Most proteins were annotated as members of the family (155); however, others were annotated as novel predictions (7). Only the longest coding sequence for each ortholog was kept.

Protein sequences for all mammalian formins were aligned with the alignment program MAFFT, v7.310 (*Katoh and Standley, 2013*) and edited in SeaView (*Gouy et al., 2010*) to build a domain phylogeny. Creation of a FH2 phylogeny was based upon previous formin family phylogenetic analyses (*Chalkia et al., 2008*). A maximum likelihood (ML) phylogenetic tree was built using the RAxML software, v8.2.9 (*Stamatakis, 2014*) with the PROTGAMMAAUTO model of substitution and 100 bootstrap replicates; the best fit model was the JTT model (*Jones et al., 1992*). The tree was mid-point-rooted using phylogenetic visualization software FigTree, v1.4.3 (*Rambaut, 2007*).

Unaligned coding sequences of mammalian formins were derived using Pal2Nal, v14 (*Suyama et al., 2006*). Coding sequence alignment was reformatted using the program trimAl, v1.3 (*Capella-Gutiérrez et al., 2009*). The FH2 mammalian tree was edited to distinguish INF clade branches as the foreground from the remaining background branches. Another FH2 phylogeny was edited distinguishing only INF2 clade branches from the rest of the tree. To examine ancient positive selection on the mammalian FH2 domain in the INF and INF2 clades, analyses of evolutionary rates were conducted using the codeml package in PAML, v4.8 (*Yang, 2007*). Branch models were used to predict the rates of codon substitution (dn/ds) based on the following hypotheses: all branches reflect equivalent rates of amino acid substitution (H0), positive selection on the INF clade (H1), and positive selection on the INF2 clade (H2).

## Cell lines

First trimester human EVT (HTR-8/SVneo) cells were cultured in RPMI-1640 medium supplemented with 5% FBS at 37°C under 5% $CO_2$ (*Graham et al., 1993*). BeWo choriocarcinoma cells, which are male in origin, were cultured in nutrient Mixture F-12 Hams medium supplemented with penicillin (100 U/mL), streptomycin (100 µg/mL), and 10% FBS at 37°C under 5% $CO_2$. HPMVECs, the sex of which are undetermined, were maintained at 37°C under 5% $CO_2$ in EGM-2 plated on Attachment Factor-coated culture flasks (*Troja et al., 2014*). BeWo choriocarcinoma cells were authenticated by STR profiling and tested negative for mycoplasma contamination by ATCC (ATCC, Manassas, VA). HTR-8/SVneo cells, a generous gift from Dr. Charles Graham at Queen's University (*Graham et al., 1993*), have also been authenticated by STR profiling and tested negative for mycoplasma contamination by ATCC. HPMVECs were isolated from normal term pregnancies under IRB approval and were tested for expression of endothelial cell markers CD31 and von Willebrand factor (*Troja et al., 2014*). Contamination by smooth muscle cells was assessed by immunocytochemistry. Mycoplasma contamination was not assessed in these cells.

## Transfection and drug treatment

HTR-8/SVneo and BeWo cells were seeded at a density of $2.5 \times 10^5$ cells per well of a 6-well plate and allowed to reach 70% confluency at the time of transfection. Transfections were performed in duplicate (triplicate for invasion assays) with 50 nM of MISSION siRNA Universal Negative Control #1 (Millipore Sigma, St. Louis, MO), 50 nM of INF2 siRNA (Thermo Fisher Scientific, Waltham, MA), or vehicle following the Lipofectamine 3000 Reagent standard protocol (Thermo Fisher Scientific). 24 hr post-transfection, the media was replaced on cells and replaced with either growth media or serum-free media. 48 hr post-transfection, cells were harvested for mRNA or protein analysis. Additional HTR-8/SVneo cells were cultured for 7 days in DMSO, 1 µM PP1 (Cayman Chemical, Ann Arbor, MI), or 1 µM TX1123 (Millipore Sigma) prior to invasion assays.

## RNA isolation and quantitative PCR

RNA was isolated from whole mouse placentas or uterus using TRIzol reagent (Thermo Fisher Scientific) per manufacturer's protocols. RNA was isolated from cells using the Qiagen RNeasy Mini Kit per the manufacturer's protocols. cDNA was synthesized from 1000 ng of RNA (Qiagen, Hilden, Germany) per manufacturer's protocols. EXPRESS SYBR GreenER (Thermo Fisher Scientific) was used for qPCR analysis. Primer sequences were generated using the NCBI primer BLAST tool for human *INF2* and mouse *Inf2* mRNAs. All mouse genes analyzed were normalized to *Ribosomal Protein S20* (*Rps20*) expression. All human genes analyzed were normalized to *Beta-Actin* (*ACTB*) expression. Gene expression data were generated and calculated using the ΔΔCt method on the StepOnePlus real time PCR system (Applied Biosystems, Foster City, CA).

## Protein isolation and analysis

Cell pellets were homogenized in RIPA Buffer (Millipore Sigma) supplemented with Protease Inhibitor Cocktail (Millipore Sigma) and Phosphatase Inhibitor Cocktails I and II (Millipore Sigma) and protein concentration determined by BCA Assay (Thermo Fisher Scientific). 30 µg was loaded per lane of a 4–12% gradient Bis-Tris polyacrylamide gel (Thermo Fisher Scientific) and transferred to a Hybond enhanced nitrocellulose membrane using a semidry transfer system (BioRad, Hercules, CA). Membranes were blocked with 5% milk in TBS with 0.1% Tween-20 (Millipore Sigma). Blots were probed with overnight at 4°C with anti-INF2 (Millipore Sigma, ABT61, 1:500) or anti-LCK (Abcam, Cambridge, UK; ab208787, 1:2000). Binding of the secondary goat anti-rabbit secondary antibody (Santa Cruz, Dallas, TX; sc-2004, 1:10,000) was determined using SuperSignal West Dura Extended Duration Substrate (Thermo Fisher Scientific). Blots were stripped with Restore Western Blot Stripping Buffer (Thermo Fisher Scientific) and reprobed with the anti-actin antibody (Millipore Sigma, 1:30,000) as an internal control.

## Matrigel invasion assay

Serum starved HTR-8/SVneo cells were plated on BioCoat Matrigel Invasion Chambers (Corning, Corning, NY) at a density of $2.0 \times 10^5$ cells in 200 µL of serum-free media. Each chamber was placed in a well of a 24-well plate containing 600 µL of RPMI medium with 10% FBS for 24 hr, after which

the Matrigel and non-invading cells were removed from the membrane. Inserts were fixed in 4% PFA and washed in PBS. Nuclei of invaded cells were stained with DAPI and membranes were mounted on slides. Invaded cells were counted in five random fields at 10x magnification in three inserts per treatment and in three independent experiments. Drug-treated HTR-8/SVneo cells were similarly tested for invasiveness after culture for 7 days in DMSO, 1 µM PP1 (Cayman Chemical), or 1 µM TX1123 (Millipore Sigma) in three independent experiments. Data are represented as percent invaded cells normalized to vehicle-treated HTR-8/SVneo cells.

## Immunocytochemistry

24 hr post-transfection, cells were transferred to a chamber slide (ibidi). 48 hr post-transfection, one subset of slides were treated for 20 min with 100 mM MitoTracker Red CMXRos (Thermo Fisher Scientific, M7512) at 37°C. Media was removed from remaining slides and cells were washed with PBS and fixed in 4% PFA. After blocking in 10% normal horse serum in 0.1% Tween-PBS, cells were incubated overnight at 4°C with MAL2 (Abcam, ab75347, 1:100) or LCK (Abcam, ab208787, 1:100). Positive staining was detected using a fluorescent donkey anti-rabbit IgG secondary antibody (Thermo Fisher Scientific, 1:200). Cells were counterstained with Alexa Fluor 594 phalloidin (Thermo Fisher Scientific, 1:500) and DAPI (Thermo Fisher Scientific, 1:10,000). Cells were washed and post-fixed with 4% PFA and were imaged on a Nikon Ti-E inverted microscope with a Nikon A1R and a 100x oil immersion objective in the Confocal Imaging Core at Cincinnati Children's. Images were processed in NIS-Elements and depicted as maximum intensity projections and are representative from three independent experiments. Imaris, v9.0.1 (Bitplane, Zurich, Switzerland), was used to determine mitochondrial volume and phalloidin content of HTR-8/SVneo cells. Cytoplasmic phalloidin content was determined by taking the ratio of cytoplasmic phalloidin volume to the total cellular phalloidin volume, which we multiplied by 100.

## Animals

*Inf2*-deficient mice (Inf2$^{tm1.1(KOMP)Vlcg}$, abbreviated *Inf2$^{-/-}$*) used for this research project were generated by the trans-NIH Knock-Out Mouse Project (KOMP) on a C57BL/6NTac background and obtained from the KOMP Repository (*Dickinson et al., 2016*). Using the ZEN-Ub1 cassette in VGB6 ES cells, 12,623 bp of *Inf2* (Chr12:112,600,006–612,628) were deleted, with an insertion of a LacZ reporter between exons 1 and 23, removing 1270 amino acids. Mice were housed on a 14-/10 hr light-dark cycle with access to chow and water *ad libitum*. Colonies were maintained as *Inf2$^{+/-}$* x *Inf2$^{+/-}$* matings, allowing for *Inf2$^{-/-}$* animals to be compared to littermate controls. Females between 10 weeks and 6 months of age were used for studies. All animal procedures were approved by the Cincinnati Children's Medical Center Animal Care and Use Committee and were in accordance with the National Institutes of Health guidelines.

## Histology and immunohistochemistry

Nulliparous females were set up with males (so pups were homozygous) for timed matings at 1700 hr and separated the following morning at 0800 hr. A copulatory plug marked 0.5 days post-coitum (dpc). Placentas at gestational days 15.5 and 18.5 were collected and fixed overnight in 4% PFA at 4°C. Tissues were washed in PBS, halved, and immersed in 70% ethanol prior to processing and paraffin embedding. Placentas were sectioned at 5 µm. Slides were baked at 60°C overnight, deparaffinized, and rehydrated. Antigen retrieval was performed using 10 mM citrate buffer (pH 6.0) followed by PBS washes. Endogenous peroxidase activity was removed by treatment with 3% $H_2O_2$ for DAB IHC. Non-specific binding was blocked by incubating slides for 1 hr in 4% normal horse serum in 0.1% Tween-PBS. Inf2 was detected using a rabbit anti-INF2 primary antibody (Millipore Sigma, HPA000724, 1:25) and using a biotinylated horse anti-rabbit IgG secondary antibody at (Vector Laboratories, Burlingame, CA; BA-1100, 1:200) followed by treatment with ABC peroxidase complex (Vector Laboratories). Slides were developed with either DAB (Vector Laboratories), counterstained with nuclear fast red and mounted with PROTOCOL Mounting Medium (Thermo Fisher Scientific), or developed with Cyanine 5 tyramide (Perkin Elmer, Waltham, MA), counterstained with DAPI (Thermo Fisher Scientific) and mounted with ProLong Gold Antifade Mountant (Thermo Fisher Scientific). Immunofluorescent co-localization of INF2 with: endothelial cells using a goat anti-Endomucin-2 antibody (R&D Systems, Minneapolis, MN; AF4666, 1:200), canal and spiral

artery-associated TGCs using a goat anti-Proliferin antibody (R&D Systems, AF1623, 1:200), and trophoblasts using Alexa Fluor 594 rabbit anti-Cytokeratin 7 (Abcam, Cambridge, UK; ab209600, 1:100). Lck localization was determined using a rabbit anti-Lck antibody (Abcam, ab3885, 1:200). Nuclei were counterstained with DAPI (Thermo Fisher Scientific, 1:10,000). Images are representative of placentas from three dams per genotype.

## Visualization of maternal circulation

At E19.0, 300 µL of 400 µg/mL DyLight 649 labeled *Lycopersicon Esculentum* (Tomato) Lectin (Vector Laboratories) in 100 U/mL heparin sulfate and PBS was injected into the tail vein. 20 min were allowed for lectin circulation. Placentas were fixed in 4% PFA overnight at 4°C. The following day, placentas were washed in PBS. Tissue was cleared via an active CLARITY technique (*Lee et al., 2016*). Placentas were incubated overnight at 4°C in a 4% acrylamide hydrogel monomer containing 0.25% Wako VA-044 photoinitiator. Tissues were incubated for 3–4 hr in a 37°C water bath to polymerize the gel. Tissues were lightly washed, then placed in a Logos Biosystems X-CLARITY machine for 6–8 hr at 1.5 Amperes and 37°C. Post clearing, tissues were washed in 37°C water overnight, then multiple washes with PBS to remove any residual SDS. Placentas were then dehydrated in a methanol series and placed in benzyl alcohol/benzyl benzoate for final clearing and mounted in a custom-made aluminum chamber slide for upright microscopy. Images were acquired on a Nikon FN1 upright microscope with a Nikon A1R-MP in single-photon confocal mode and a 16X/0.8 NA water immersion objective in the Confocal Imaging Core at Cincinnati Children's. Images were stitched, analyzed and processed in NIS-Elements. A researcher blinded to mouse genotype counted and recorded number of spiral arteries. Images and videos are representative.

## Systolic blood pressure measurements

Systolic blood pressure was measured in conscious animals using a non-invasive Volume Pressure Recording method (Kent Scientific, Torrington, CT), a previously validated method (*Feng et al., 2008*). Females were trained in the restraints on the warming platform for two weeks prior to the study. Baseline measurements were an average of measurements on virgin females for three consecutive days. Plugged females resumed training at E7.5 to avoid interfering with implantation while non-plugged females resumed training until plugged. Measurements were taken longitudinally from E14.5 through postnatal day 7. Three cycles used to acclimate females to tail cuff inflation and discarded from analysis, using acceptable reads from the following 12 cycles. Investigator observed all measurement cycles and manually discarded reads with signal artifacts. Data collected by Kent Scientific software were analyzed off-line. All blood pressure measurements were obtained between 0700 and 1000 hr.

## Circulating angiogenic factors

Maternal blood was collected in serum separator tubes at E15.5 and E18.5 by submandibular phlebotomy. Blood was allowed to clot at room temperature for 30 min and serum was removed and stored per the manufacturer's conditions until assayed by the Research Flow Cytometry Core at CCHMC on a MILLIPLEX MAP Mouse Angiogenesis/Growth Factor Magnetic Bead Panel and MILLIPLEX MAP Mouse Soluble Cytokine Receptor Magnetic Bead Panel (Millipore Sigma).

## Parturition and pregnancy outcome

Nulliparous $Inf2^{-/-}$ or $Inf2^{+/+}$ females were set up with homozygous males of corresponding genotype for timed matings at 1700 hr and separated the following morning at 0800 hr. Visualization of a copulatory plug marked 0.5 days post-coitum (dpc). Cages were checked four times daily (0600, 1000, 1400, 1800 hr) to calculate gestation length, determined by the timing of birth of the first pup. Number of pups in a litter and pup weights were recorded at birth (P0) or at E18.5 after euthanization. Serum progesterone concentration at E18.5 (by submandibular phlebotomy as described above) was measured by ELISA according to manufacturer's protocols (BioVendor, Brno, Czech Republic; RTC008R). $PGF_{2\alpha}$ and $PGE_2$ were measured on snap frozen uterus at E18.5. Prostaglandins were extracted after weighing frozen tissue by homogenization in 100% ethanol. Centrifugation removed debris and the supernatant dried down under inert gas, resuspended and assayed per the manufacturer's instruction (Oxford Biomedical Research, Rochester Hills, MI).

## Transabdominal umbilical Dopplers

A Vevo 2100 ultrasound machine (Fugifilm VisualSonics, Toronto, Canada) equipped with a 40 MHz transducer was used to perform fetal ultrasounds on E18.5. Dams were anesthetized with 1.0% inhaled isoflurane, abdominal hair removed with a depilatory agent, and positioned on a warmed platform to maintain euthermia. After fetal number and placement were determined, each fetus was examined consecutively around the uterine horn. Seven fetuses per litter were scanned in each dam. Umbilical vessels were identified using 2-dimensional and color Doppler imaging with the vessels traced from the fetus to the site of insertion into the placenta. A freely mobile loop of umbilical cord was interrogated. Color Doppler images of the umbilical artery and vein were recorded, typically capturing flow in both vessels simultaneously. The pulsed wave Doppler sample volume was adjusted and subtle positional changes of the transducer made to obtain umbilical vessel interrogation as close to parallel flow as possible, adjusting the beam angle from 0 to 60 degrees as needed to provide the best alignment. Pulsed wave Doppler is recorded at a sweep speed of 5.1 m/second with peak velocity scaled to a maximum of 150 mm/second to optimize tracings for off line analysis. Images were analyzed using the vascular package included in the Vevo 2100 software by investigators blinded to genotype. Off line measurements include fetal heart rate, umbilical artery peak systolic velocity (PSV), end diastolic (EDV) and the umbilical artery velocity time integral (VTI), the latter of which provides the umbilical artery mean velocity. Umbilical vein flow was qualitatively assessed for abnormalities such as pulsatile diastolic flow or flow reversal. Images highlight the largest differences.

## Histomorphometry

Vascular density was measured in placentas at E15.5 and E18.5 by counting the number of blood vessels, identified using IF for Endomucin, per high-powered field (40X magnification), blinded to genotype. Vessel numbers in each of 10 random fields were averaged in a single section per placenta across 2–3 placentas per dam. An average of three measurements made across 20X H&E scans of each the labyrinth, junctional zone, and whole placenta were taken using the Nikon Elements Software. Results were reported from 2 to 3 placentas per dam at E18.5. Images are representative.

## CTB/Endothelial cell crosstalk

24 hr post-transfection, media on transfected BeWo cells was replaced with EGM-2 (Lonza, Basel, Switzerland). 48 hr post-transfection, BeWo cells were harvested for RNA analysis while the media was removed and placed on HPMVECs at 70% confluence grown in 6-well plates coated with attachment factor (Thermo Fisher Scientific). In a subset of BeWo cells, media and cells were collected for PGF ELISA analysis (R&D Systems, DPG00). ELISA data were normalized to total protein. HPMVECs were cultured in BeWo cell conditioned medium for 48 hr prior to harvesting for RNA analysis.

## Statistics

Data were analyzed by Student's $t$ test, one-, or two-way ANOVA test (Prism 7.0c software; Graph-Pad Software, Inc., San Diego, CA) as indicated in Figure Legends and a $p \leq 0.05$ was considered significant. The n represents either dams or fetuses as indicated in Figure Legends. Results are reported as ±SEM.

## Acknowledgements

This work was supported by NIH grant S10RR029406 for the Nikon A1R Upright Multiphoton housed in the CIC at Cincinnati Children's and the March of Dimes Prematurity Research Center Ohio Collaborative. MLJ is a Littlejohn Summer Research Scholar at Vanderbilt University. The mouse strain used for this research project was generated by the trans-NIH Knockout-Mouse Project (KOMP) and obtained from the KOMP repository. We thank Dr. Matt Kofron and the CIC at CCHMC for their assistance with imaging, the CICRL at CCHMC for their help with the fetal Dopplers, Dr. Jesse Slone and Dr. Taosheng Huang for assistance with the mitochondria studies, and Dr. CH Graham (Queen's University, Canada) for providing the HTR-8/SVneo cell line. The authors have declared that no conflict of interest exists.

## Additional information

### Competing interests
Antonis Rokas: Reviewing editor, *eLife*. The other authors declare that no competing interests exist.

### Funding

| Funder | Grant reference number | Author |
| --- | --- | --- |
| March of Dimes Foundation | 22-FY16-125 | Katherine Young Bezold Lamm<br>Maddison L Johnson<br>Julie Baker Phillips<br>Antonis Rokas<br>Louis J Muglia |

The funders had no role in study design, data collection and interpretation, or the decision to submit the work for publication.

### Author contributions
Katherine Young Bezold Lamm, Conceptualization, Data curation, Formal analysis, Validation, Investigation, Visualization, Methodology, Writing—original draft, Writing—review and editing; Maddison L Johnson, Helen N Jones, Data curation, Methodology, Writing—original draft, Writing—review and editing; Julie Baker Phillips, Data curation, Writing—review and editing; Michael B Muntifering, Visualization, Methodology, Writing—original draft; Jeanne M James, Data curation, Writing—original draft, Writing—review and editing; Raymond W Redline, Methodology, Writing—review and editing; Antonis Rokas, Writing—original draft, Writing—review and editing; Louis J Muglia, Conceptualization, Supervision, Funding acquisition, Writing—review and editing

### Author ORCIDs
Katherine Young Bezold Lamm (iD) http://orcid.org/0000-0001-9803-5743
Michael B Muntifering (iD) http://orcid.org/0000-0002-0460-4050
Antonis Rokas (iD) https://orcid.org/0000-0002-7248-6551

### Ethics
Animal experimentation: All animal procedures were approved by the Cincinnati Children's Medical Center Animal Care and Use Committee (Protocols #2013-0097, #2013-0111, and #2017-0051) and were in accordance with the National Institutes of Health guidelines.

### Decision letter and Author response
Decision letter https://doi.org/10.7554/eLife.31150.024
Author response https://doi.org/10.7554/eLife.31150.025

## Additional files

### Supplementary files
• Transparent reporting form
DOI: https://doi.org/10.7554/eLife.31150.022

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
