## [Decision Letter]

Thank you for submitting your article "Inverted Formin-2 regulates intracellular trafficking, placentation, and pregnancy outcome" for consideration by *eLife*. Your article has been favorably evaluated by Harry Dietz (Senior Editor) and three reviewers, one of whom is a member of our Board of Reviewing Editors. The reviewers have opted to remain anonymous.

The reviewers have discussed the reviews with one another and the Reviewing Editor has drafted this decision to help you prepare a revised submission.

Summary:

The authors tested the hypothesis that inverted formin 2 (*Inf2*) plays an important role in placentation, including trophoblast invasion and remodeling of the uterine wall, i.e., formation of the maternal-fetal interface. This is in keeping with the fact that the placenta is one of the sites where this molecule is expressed and the functions of INF2 include severing and polymerizing actin filaments as well as microtubule bundling, important for motility and invasion. The authors use a combination of human cell and mouse knockout models to explore the role of *Inf2* in trophoblast invasion of the uterus, a cancer-like process, and attendant remodeling of the spiral arteries, vessels that traverse the uterine lining. First, they used a human cell line with trophoblast-like properties (HTR-8/SVneo cells) to show that knockdown of INF2 restricted LCK localization to the nuclear region of the cells; in controls, this molecule was distributed throughout the cytoplasm. In pregnant mice, expression of *Inf2* peaks on gestational day 15.5 and localizes to the trophoblast cells of control but not *Inf2^-/-^* placentas. Lck distribution was described as distributed throughout the cytoplasm of trophoblast cells in control animals and restricted to the peri-nuclear region of trophoblast cells in null animals. At a physiological level, blood pressure was significantly elevated at gestational day 17.5 in *Inf2^-/-^* mice as compared to controls, returning to normal after birth. Other endpoints were also impacted in the null animals. Gestation length was increased by 9.8 hours. Fetal weight on gestational day 18.5 was significantly reduced, but normal at birth, suggesting catch up growth. Fetal vascular density in the placental labyrinth was significantly higher in *Inf2^-/-^* pregnancies and the depth of this region was decreased. In summary, manuscript demonstrates a new mouse model of placental vasculopathy that recapitulates some of the phenotypes of impaired placentation. Placentation field has lacked good animal models; this manuscript would be a valuable contribution. We have the following suggestions to improve the manuscript.

Essential revisions:

1) Knockdown of INF2 significantly increased placental growth factor mRNA expression and culture medium from the cells increased the expression of soluble VEGFR1 mRNA by human placental microvascular endothelial cells. These experiments would be strengthened by the use of primary human trophoblasts and analyses of angiogenic factors at the protein level.

2) Pregnancy phenotypes appear relatively mild. A better characterization of the phenotypes would greatly enhance this paper. For example,

a) Figure 4, placental weights are missing; it would be nice to show placental/fetal ratio to evaluate if there is true placental insufficiency.

b) Birth weights are not changed at P0; – this does not appear like severe IUGR. Can the discrepancy in the birth weights at E18.5 and PO be explained by latter gestational age of delivery. Most preeclamptics with severe fetal growth restriction would have preterm delivery, which is not seen in this mouse model.

c) It would be good to report total litter weight in addition to litter size and fetal weight.

d) Proteinuria or other features of preeclampsia such as thrombocytopenia or elevated LFTs would be informative. At best the paper as presented is a model of gestational hypertension with some features of placental insufficiency.

3) Additionally, the authors do not account for other well-known functions of INF2, including a role in mitochondrial fission. Did this play a part in the phenotype of the placentas of *Inf2^-/-^* mice? Also, possible actin and microtubule effects were not explored but should be investigated and discussed.

---

## [Author Response]

Essential revisions:1) Knockdown of INF2 significantly increased placental growth factor mRNA expression and culture medium from the cells increased the expression of soluble VEGFR1 mRNA by human placental microvascular endothelial cells. These experiments would be strengthened by the use of primary human trophoblasts and analyses of angiogenic factors at the protein level.

We agree with the reviewers that these experiments would be strengthened by the use of primary human cytotrophoblasts and analysis of angiogenic factors at the protein level. Our original experimental design involved using freshly isolated primary human trophoblasts and knockdown of *INF2* using a nanoparticle delivery method. However, we were concerned that the timing of cell and media collection 48 hours after siRNA delivery (52 hours after initial plating) would yield a mixed population of cytotrophoblasts and syncytiotrophoblasts. However, we proceeded with the suggested experiments using primary human trophoblasts with the anticipation that adequate replication of our conditioned media experiments would not be possible as primary human cytotrophoblasts do not survive in human placental microvascular endothelial cell media as the BeWo cells do. Using a nanoparticle delivery method, we achieved 65% knockdown of *INF2* mRNA in primary cytotrophoblasts with no change in relative expression of nonsense siRNA-treated cells compared to untreated cells (n = 4). hCGβ ELISA data confirmed the presence of mixed cell populations that greatly varied from experiment to experiment. As we are interested in the ability of INF2 to regulate cytotrophoblast-fetal endothelial cell crosstalk, the inability to control for the presence of syncytiotrophoblasts unfortunately renders this system insufficient to answer this question.

As knockdown of *INF2* significantly increased *PlGF* mRNA in the BeWo cell line and HPMVEC exposure to cultured media from these cells significantly increased *sFLT1* mRNA in response, we hypothesized PlGF secretion would also be increased and underlie the *sFLT1* response in the HPMVECs. PlGF secreted by *INF2* siRNA-treated cells was significantly higher than PlGF secreted by vehicle-treated cells (Figure 7= 0.037). As HPMVECs are treated with conditioned media, measuring angiogenic factor secretion from these cells would be unfeasible as it would be difficult to parse which angiogenic factors came from the HPMVECs versus the factors secreted by the BeWo cells. As our prime interests lie in the ability of *INF2* to regulate angiogenic factor expression and secretion, we believe the PlGF ELISA data are satisfactory to answer this question and are sufficient to effect changes on downstream target cell types. However, further studies to determine the precise regulatory mechanisms by which INF2 modulates expression and secretion of angiogenic factors in trophoblasts are required.

2) Pregnancy phenotypes appear relatively mild. A better characterization of the phenotypes would greatly enhance this paper. For example,a) Figure 4, placental weights are missing; it would be nice to show placental/fetal ratio to evaluate if there is true placental insufficiency.

To better characterize the pregnancy phenotypes of *Inf2^-/-^* mice, we have added placental weights (which are not significantly altered) to Figure 5. We have also added the fetal:placental weight ratio (which is significantly reduced in *Inf2^-/-^* animals at E18.5; *p* = 0.0190)to Figure 5. This finding suggests the presence of true placental insufficiency, which we have added to the Discussion.

b) Birth weights are not changed at P0; – this does not appear like severe IUGR. Can the discrepancy in the birth weights at E18.5 and PO be explained by latter gestational age of delivery. Most preeclamptics with severe fetal growth restriction would have preterm delivery, which is not seen in this mouse model.

To the Discussion, we have added the hypothesis that the growth restricted *Inf2^-/-^* fetuses do not differ in weight at time of birth from wildtype pups due to the increased gestation length in knockout animals. We have also added in the Discussion that women with PE and severe IUGR commonly deliver preterm due to clinical intervention to avoid maternal and fetal demise; therefore, it is unknown whether the length of gestation would be increased in these women as it is in our mouse model to allow for continued growth.

c) It would be good to report total litter weight in addition to litter size and fetal weight.

We have added the total litter weight (Figure 5—figure supplement 1) to our manuscript.

d) Proteinuria or other features of preeclampsia such as thrombocytopenia or elevated LFTs would be informative. At best the paper as presented is a model of gestational hypertension with some features of placental insufficiency.

Despite trends in the right direction, we believe the differences in total litter weight are not significantly different due to confounding by variability in the number of pups per litter. Although we did not detect proteinuria in these mice, proteinuria may be masked in our transgenic line as C57BL/6 mice are resistant to developing proteinuria (Ishola DA et al. Nephrol Dial Transplant, 2006). Together, our data suggest the *Inf2* knockout mouse is a model of placental insufficiency with features of gestational hypertension and intrauterine growth restriction.

3) Additionally, the authors do not account for other well-known functions of INF2, including a role in mitochondrial fission. Did this play a part in the phenotype of the placentas of Inf2^-/-^ mice? Also, possible actin and microtubule effects were not explored but should be investigated and discussed.

Consistent with previous studies, loss of *INF2* significantly increased mitochondrial size in vitro (Korobova F et al. Science, 2013). Our data, therefore, supports a role for INF2 in mitochondrial fission. Studies suppressing dynamin-related protein 1 (Drp1) – a fission protein whose recruitment to mitochondria is facilitated by INF2 (Jin X et al. PLoS Genet, 2017; Korobova F et al. Science, 2013) – in breast cancer cells inhibited formation of lamellipodia and suppressed their migration and invasion capabilities (Zhao J et al. Oncogene, 2013). However, as knockdown of Drp1 results in a significantly more severe mitochondrial phenotype, we support the suggestion put forth by Korobova et al. that there are INF2-independent pathways in mitochondrial fission. Therefore, we surmise that INF2-dependent trophoblast invasion seen in our study is not only mitochondrial-dependent.